# VLMbench: A Compositional Benchmark for Vision-and-Language Manipulation

**Kaizhi Zheng**
University of California, Santa Cruz
kzheng31@ucsc.edu

**Xiaotong Chen**
University of Michigan, Ann Arbor
cxt@umich.edu

**Odest Chadwicke Jenkins**
University of Michigan, Ann Arbor
ocj@umich.edu

**Xin Eric Wang**
University of California, Santa Cruz
xwang366@ucsc.edu

## Abstract

Benefiting from language flexibility and compositionality, humans naturally intend to use language to command an embodied agent for complex tasks such as navigation and object manipulation. In this work, we aim to fill the blank of the last mile of embodied agents—object manipulation by following human guidance, e.g., "move the red mug next to the box while keeping it upright." To this end, we introduce an Automatic Manipulation Solver (AMSolver) system and build a Vision-and-Language Manipulation benchmark (VLMbench) based on it, containing various language instructions on categorized robotic manipulation tasks. Specifically, modular rule-based task templates are created to automatically generate robot demonstrations with language instructions, consisting of diverse object shapes and appearances, action types, and motion constraints. We also develop a keypoint-based model 6D-CLIPort to deal with multi-view observations and language input and output a sequence of 6 degrees of freedom (DoF) actions. We hope the new simulator and benchmark will facilitate future research on language-guided robotic manipulation. Project website: https://sites.google.com/ucsc.edu/vlmbench/home.

## 1 Introduction

"Can you help me to clean the disks in the sink?" — humans communicate with each other using language to issue tasks and specify the requirements. Although recent progress in embodied AI pushes intelligent robotic systems to reality closer than at any other time before, it is still an open question how the agent learns to manipulate objects following language instructions. Therefore, we introduce the Vision-and-Language Manipulation (VLM) task, where the agent must follow language instructions to do robotic manipulation. Recent benchmarks were developed to evaluate robotic manipulation tasks with language guidance and visual input [11, 1, 36]. However, the collected task demonstrations are not modular and can hardly scale because they lack (1) adaptation to novel objects and (2) categorization for modular and flexible composition to complex tasks. Additionally, the lack of variations in language also leads to biases in visual reasoning learning. To deal with these problems, we expect an inclusive, modular, and scalable benchmark to evaluate embodied agents for various language-guided manipulation tasks.

An ideal VLM benchmark should have at least three characteristics: The first one is scalability. Such a benchmark should automatically generate various physics-realistic 6 degrees of freedom (DoF) interactions with affordable objects and expand new tasks effortlessly. The second is task categorization, which exploits commonality concerning robot motion between semantic tasks and is

36th Conference on Neural Information Processing Systems (NeurIPS 2022) Track on Datasets and Benchmarks.

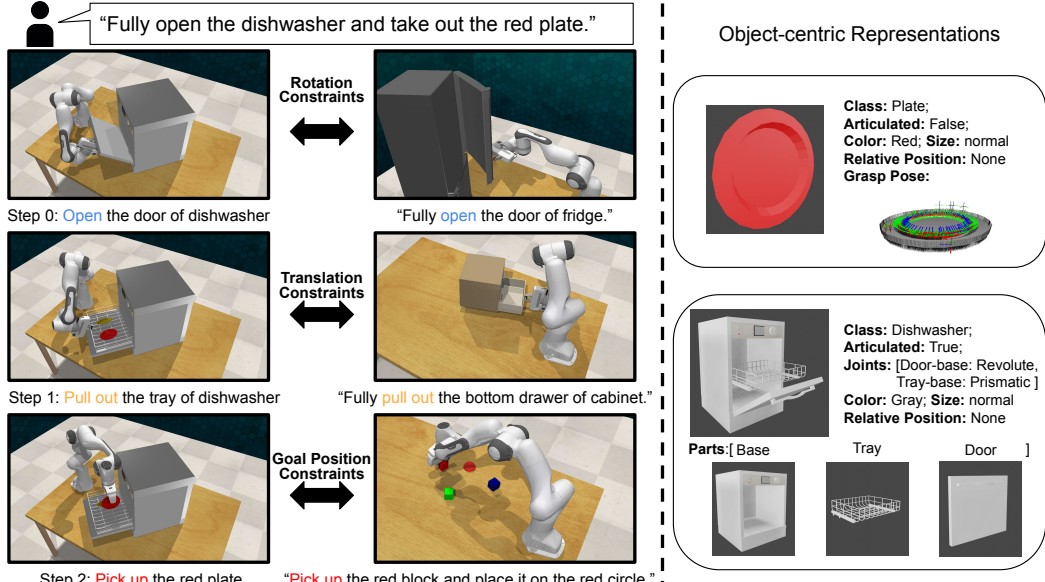

Figure 1: Given the language instructions and observations, the VLMbench requires the agent to generate an executable manipulation trajectory for specific task goals. On the left, we show that the complex tasks can be divided into unit tasks according to the constraints of the end-effector, like "Open the door of the dishwasher" and "Open the door of the fridge" should both follow the rotation constraints of the revolute joint. On the right, we show examples of object-centric representations, where all graspable objects or parts generate local grasping poses as their attributes. Depending on the modular design, we can generate reasonable VLM data automatically.

almost ignored in existing works. The third is reasonable language generation, which requires the benchmark to generate language instructions for testing diverse visual reasoning abilities without biases. However, existing benchmarks [30, 11, 36, 1] lack at least one characteristic for VLM tasks. Motivated by these attributes, we present VLMbench, a highly categorical robotic manipulation benchmark with compositional language for visual reasoning. To build and scale VLMbench, we propose AMSolver, an automatic unit task builder that can compose unit tasks to create complex multi-step tasks and seamlessly adapt to novel objects. Compared to previous benchmarks, VLMbench categorizes manipulation tasks into various meta-manipulation actions according to the constraints of robot trajectories for the first time. Meanwhile, the combinations of compositional language templates and object-centric representations provide numerous variations for visual reasoning in VLMbench, as shown in Figure 1.

To investigate the difficulty of the benchmark, we test them with several partially modal methods and a keypoint-based method, 6D-CLIPort, modified from the state-of-the-art language-guided manipulation method CLIPort [25]. The results show that there is still a massive room for improvement in the robust manipulation action generations and accurate language-guided visual understanding. To sum up, our contributions to this work include the following:

- AMSolver, an automatic demonstration generator for various task semantics, motion constraints, object types, and states defined in a novel task template formulation.

- VLMbench, a robot manipulation benchmark on 3D tasks with visual observation and compositional language instructions, where we categorize the manipulation tasks by constraints and provide variations with minimal biases in the first time.

- 6D-CLIPort, a general vision-and-language manipulation baseline model evaluated on all kinds of VLMbench tasks.

| Benchmark | novel object adaptation | automatic trajectory generation | variant object property | automatic 6-DoF grasping | constraint-based task formulation |
|---|---|---|---|---|---|
| CausalWorld [1] | ✗ | ✗ | ✓ | ✗ | ✗ |
| MetaWorld [36] | ✗ | ✓ | ✗ | ✗ | ✗ |
| ManipulaTHOR [6] | ✗ | ✗ | ✓ | ✗ | ✗ |
| Habitat 2.0 [30] | ✓ | ✓ | ✓ | ✗ | ✗ |
| Robomimic [16] | ✗ | ✗ | ✗ | ✗ | ✗ |
| BEHAVIOR [27] | ✗ | ✓ | ✓ | ✗ | ✗ |
| RLBench [11] | ✗ | ✓ | ✓ | ✗ | ✗ |
| CALVIN [18] | ✗ | ✗ | ✗ | ✗ | ✗ |
| VLMbench (ours) | ✓ | ✓ | ✓ | ✓ | ✓ |

Table 1: Comparison of existing robotic manipulation benchmarks to VLMbench.

## 2 Related Work

**Robotic Manipulation Benchmarks** There are plenty of benchmarks proposed related to visual-language robotic tasks. ALFRED [26] was proposed to do virtual object rearrangement tasks guided by visual observation and language instruction between different room-scale locations. ManipulaTHOR [6] introduced realistic interaction with objects using a 6 DoF configurable mobile manipulator. Habitat 2.0 [30] and BEHAVIOR [27] incorporated real robot navigation with simple object interaction implementation for more comprehensive mobile manipulation tasks. CALVIN [18] collects 24 hours of playing data with natural language instructions for long-horizon manipulation tasks. Regarding static manipulation benchmarks, MetaWorld [36] collects a series of translation-only tasks with demonstrations for reinforcement learning. CausalWorld [1] focused on causal inference within manipulation and showed examples of several simple object rearrangement tasks. RLBench [11] collected 100 different tasks by specifying robot arm end-effector waypoints for each of them for robot learning. Besides, crowd-sourcing platforms collect human demonstrations of a variety of common manipulation tasks through VR/AR devices, such as Robosuite [42] and Robomimic [16]. Compared to these works, our VLMbench includes high-level task descriptions and low-level robot action translations and realizes automatic task builders for easier generation of complex tasks.

**Vision and Language Tasks for Embodied AI** Vision and language tasks, such as Vision Question Answering (VQA) [3, 5], Video Captioning [33, 7, 31], Image-Text Retrieval [40, 4], connect vision and NLP research to produce semantics from combined embedding features. Recent works such as referring object spatial relations [13] and dynamic events [35] provide valuable reasoning results for the agent to act. Vision-and-Language Navigation (VLN) [2, 32, 34, 21, 20, 9, 41] proceeds another step to use visual observations and language instructions for robotic navigation directly. Recent benchmarks [26, 30, 27] combined VLN with abstracted object interaction to include more object rearrangement tasks. Compared to the tasks mentioned above, our benchmark is designed for more complex robotic manipulation that simulates physics-realistic 6DoF grasping and requires reasoning ability from the visual-language space to the executable action space.

**Language-Instructed Manipulation** Various manipulation tasks have been recently researched with language input, either describing the entire task or serving interactive input for task specifications. HULC [17] proposes a hierarchical network for long-horizon manipulation tasks, which contains multi-modal transformers for task planning and a policy network for action generation. Structformer [14] proposes an object selection network from language and visual encodings and a language-conditioned pose generator for semantic object rearrangement. Stepputtis et al. [28] proposed a closed-loop control model for pouring tasks. CLIPort [25] proposed a two-stream framework to learn a spatial attention map for 2D object manipulations. Lynch et al. [15] fed natural language instructions to a goal-conditioned policy pretrained using imitation learning for various tasks. INVIGORATE [39] proposed an interactive system that takes language input to correct false estimations and re-plan for object grasping in clutter. Shao et al. [24] trained a joint language-vision model on 7-DoF goal trajectory estimation with another task classifier for multi-task training. Goyal et al. [8] learned zero-shot task adaptation to accomplish novel tasks through differences described in natural language. In this paper, our 6D-CLIPort model learns full 6D grasping and conducts variate-length robot manipulation tasks from language instructions.

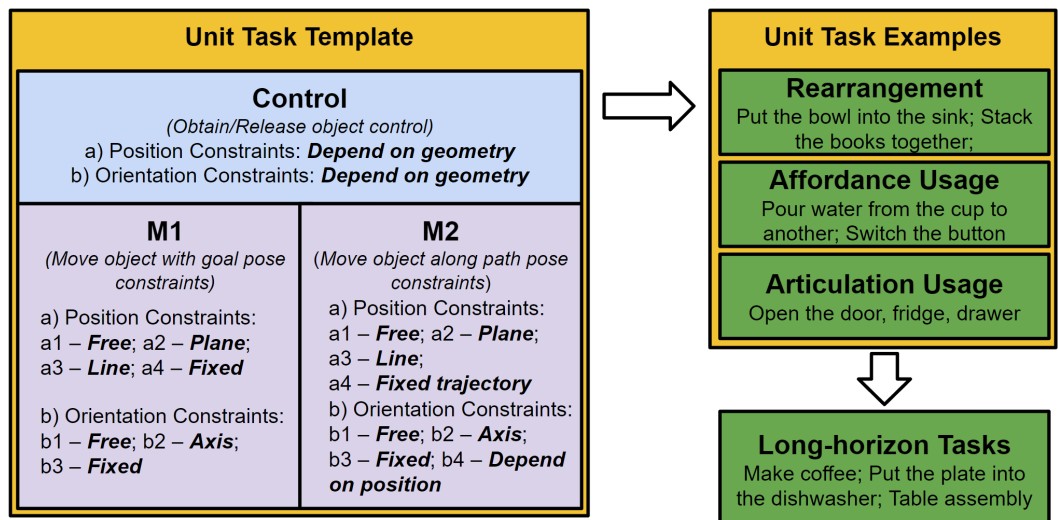

Figure 2: The unit task templates of AMSolver. On the left, we show three unit task templates parameterized by position and orientation constraints over the robot end-effector on either the goal pose or the entire path. By combining these unit task templates, various task examples can be generated. For example, on the right, we show three main common task types household tasks and long-horizon tasks composed of unit tasks.

## 3 AMSolver: Automatic Manipulation Solver

We consider a rule-based task categorization that supports most daily manipulation tasks, which follows a unified formulation. To this extent, we propose Automatic Manipulation Solver (AMSolver)[1]. We focus on simple unit tasks and introduce a unit task template that categorizes motion constraints and generates a wide range of household tasks (see examples in Figure 2). The formulation treats objects as a representation that could have variations in appearance and enables automatic demonstration generation using off-the-shelf tasks and motion planners.

### 3.1 Rule-based Unit Tasks

Since the tasks have notable variations, we assume that each complex task can be decomposed into combinations of several unit tasks from the aspects of end-effector trajectories. We define a *unit task* as the semantic step of completing a sub-goal of the entire task. Specifically, a unit task is defined in a formula of 'take action on an object under certain constraints' where two constraints can parameterize a unit task: (a) **position constraints** and (b) **orientation constraints**, which describe the valid spatial space or orientation range, respectively, of the end-effector for a specific task. We propose three unit task templates detailed below that can compose the aforementioned complex tasks.

**(1) Control** is a preparation or ending step of a task, which models the transition of the object state, where the state indicates whether the robot can move the object or not. In this work, we specify one way of transition: to obtain control by grasping it and release control by opening the gripper. There are other ways to obtain control, like pushing, hitting, etc. However, these transitions will naturally lead to the constraints in the following sub-tasks, so they cannot be considered a general component for any complex task. In this unit task, the position and orientation constraints depend on the geometry of object instances.

**(2) M1** denotes moving the target object with goal pose constraints, which can be modeled as a 6 DoF transform in the robot's workspace. The position constraints define a bounded goal space in $\mathbb{R}^3$, while the orientation constraints define a valid 3D orientation $SO(3)$. We consider four types of position constraints: (a1) *Free*, (a2) *Plane*, (a3) *Line*, and (a4) *Fixed*, which means the goal position is any point inside a 3D space (a1), constrained in a 2D plane (a2), constrained in a line-shape area (a3), or fixed to a certain point (a4). There are three types of orientation constraints: (b1) *Free*, (b2)

---
[1]AMSolver is implemented in CoppeliaSim [23] (Free Educational License) and codes are based on RLbench [11] (RLBench Software License) and PyRep [10] (MIT License).

*Axis*, and (b3) *Fixed*, which means the goal orientation is unlimited in 3D rotation space (b1), only rotated along an axis in space (b2), or fixed at a given orientation (b3). For example, M1[a2, b1] can represent placing the object on a tabletop (with plane position constraint), while M1[a3, b2] can represent moving the object to a position on one line and ending with a constrained orientation of one axis, like dropping a stick into the hole.

**(3) M2** denotes moving the target object along a trajectory while satisfying the motion constraints during the entire path, which implies a more strict condition than **M1**. The constraints are mostly from object articulation, like revolute joints on doors, or task-specific requirements, like keeping the opening upward for a full-filled mug. Therefore, there are four kinds of position constraints: (a1) *Free*, (a2) *Plane*, (a3) *Line*, and (a4) *Fixed trajectory*, which means any position inside a space (a1), a plane (a2), a line (a3) or a predefined path (a4) should be feasible for the trajectory, and four kinds of orientation constraints: (b1) *Free*, (b2) *Axis*, (b3) *Fixed*, and (b4) *Depend on position*, which means every pose in the trajectory should be unlimited orientations (b1), at most rotated along an axis (b2), fixed to a certain orientation (b3), or depended on the corresponding positions. For example, M2[a2, b2] means moving the object inside a 2D plane while maintaining the orientation of one axis, like wiping the table. In contrast, M2[a4, b2] means moving the object along a fixed trajectory with maintaining the orientation of one axis, like using a screwdriver to tighten a screw. It is worth mentioning that M2[a1,b1] will degenerate to M1[a1,b1]. According to our unit task definition, we have covered every feasible 6 DoF pose of the end-effector. Therefore, we have reason to believe that these unit tasks can represent any complex task in the action space.

## 3.2 Object-centric Representation

Some recent works [12, 37] have used object-centric representations for manipulation. Since the properties are defined on objects, these representations can easily cross the variations of environments, agents, and tasks. Our benchmark assumes that objects used in the tasks are rigid and their fundamental properties will not change during the tasks. Therefore, we can parameterize the objects as a set of configurations, including class, color, size, and geometry shape. If the object is articulated, its whole configuration will contain the configuration of each part and the physical constraint of each connection. For example, a door consists of three parts: the door base, plank, and handle, so its configuration will contain each part's configuration and record the positions and ranges of two revolute joints.

## 3.3 Automatic Demonstration Generation

To create a task example, the user could compose a task template from the unit library of formulation, such as a unit task of object rearrangement as **Control**-**M1** or a more complex task of object stacking as **Control**-**M1**-**Control**-**M1**-**Control**-**M1**, etc., and then select the objects from a given set to be included in the scene as the object to be manipulated and distractor objects. The object placement could be randomized to the customer's specifications in the simulation. The corresponding language descriptions could also be generated from templates.

To implement **Control** as grasping, we create an object-wise grasp pose dictionary. Specifically, a point cloud-based grasping pose detection algorithm [19] is implemented to search all feasible grasping poses, given the object's shape and robot gripper parameters. The grasp poses are saved and transformed to world space for a particular task by simply multiplying with the object's pose. To implement **M1** and **M2**, we integrate customized motion constraints in the OMPL motion planner library so that the calculated trajectory satisfies the constraints automatically. Please refer to Appendix B for more details and task examples.

# 4 VLMbench: Visual-and-Language Manipulation Benchmark

## 4.1 Problem Definition

Given language instructions, the Vision-and-Language Manipulation (VLM) task requires an embodied agent to follow the instructions to complete tabletop manipulation tasks. Formally, at the beginning of the task, the agent receives a set of language instructions $L = \{L_1, L_2, ..., L_n\}$, where $L_i$ denotes one sentence of arbitrary length. The initial state $s_0$ contains multi-view RGB images, depth images, segmentation information, and robot states, including joint angles, velocities and torques, and

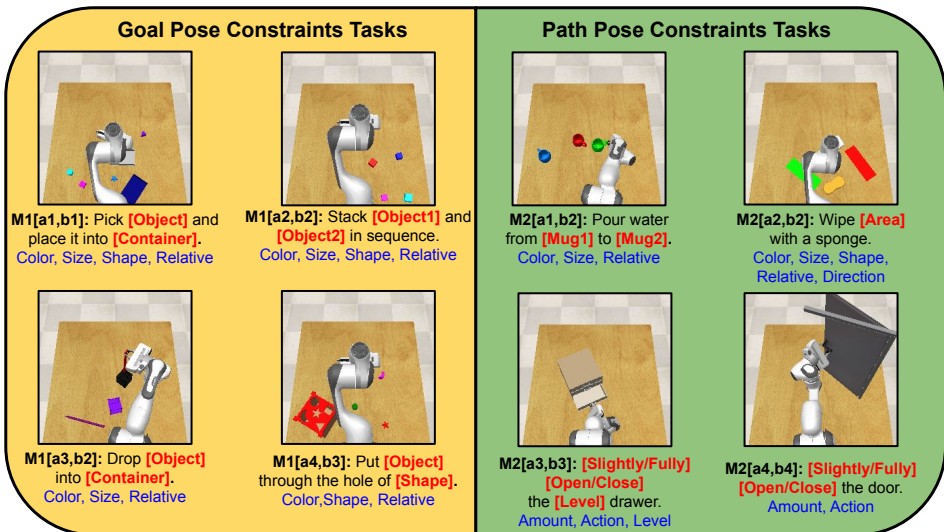

Figure 3: We show four task categories with goal-pose constraints on the left side, and on the right side, we show four task categories with path-pose constraints. For each task category, we visualize the observations of the overhead view and list the main unit task, variations, and instruction templates. The red words with brackets indicate the blank in which the variations descriptions can fill. The blue words indicate the variations in the tasks. The combinations of variations will lead to various instance-level tasks.

end-effector pose. Given the observations and language instructions, the agent needs to estimate an executable action command $a_0$, directly working on the end-effector or joints. Then, at each step $t$, the agent receives new observations $o_t$ and generates the action $a_t = f(s_t|s_0, s_1, ..., s_{t-1}, L)$ for the next step. The step loop will repeat until the agent sends a stop action or should be terminated, e.g., achieve the success conditions or the limitation steps. The agent should obey the constraints provided by language instructions during the run.

### 4.2 Tasks and Dataset

In previous works, the researchers manually designed manipulation tasks by implicit prior knowledge without categories. Instead, we are trying to build tasks from the perspectives of elementary manipulation abilities. In other words, since different tasks have various semantic meanings, we consider the task with the same unit tasks combination should be in the same category from the aspect of the action space. For example, "Open the door of the fridge," and "Open the door of the microwave" require the same action ability except for semantic meanings and grasping poses which depend on the object geometry. Therefore, we define eight general task categories, represented by one typical task in each category, shown in Table 4.2 and Fig. 3. We use the definitions in the unit task templates to represent the main constraints of each task category. The task details can be found in Appendix A, and dataset statistics can be found in Appendix C.

**Task Variations** The manipulated object's properties can randomly change for each task category, and every combination leads to a task instance. In the VLMbench, we use eight variations: color, size, relative position, shape, direction, level, amount, and action type. The variations are from two perspectives: object and motion. Object variations include color, size, shape, relative position, and direction. Here, the **color** is chosen from 20 colors in the seen settings. The **size** contains the relative descriptions between two objects, "smaller" and "larger", and descriptions between three objects, which are "large","medium", and "small". The **shape** contains five types of objects for the seen and unseen settings. The **relative position** describes the spatial relationship between two objects, like the top, front, rear, left, and right. The **direction** contains two descriptions for rectangular prism in a plane:"horizontal" and "vertical". For the objects that have the vertical structure, the **level** includes "top", "middle," and "bottom." From the motion view, the variations are amount and action type. The **amount** means how far the task needs to be done, consisting of "fully" and "slightly." The **action**

| Task Categories | Main Constraints | Variations | Instructions Samples |
|---|---|---|---|
| Pick & Place objects | M1 – Position: Free, Orientation: Free | Color, Size, Shape, Relative Position | "Pick the red cube and place it into the green container." |
| Stack objects | M1 – Position: Plane, Orientation: Axis | Color, Size, Shape, Relative Position | "Stack the small star and the medium star in sequence." |
| Drop pencil | M1 – Position: Line, Orientation: Axis | Color, Size, Relative Position | "Drop the left pencil into the right container." |
| Put into shape sorter | M1 – Position: Fixed, Orientation: Fixed | Color, Shape, Relative Position | "Put the triangular prism through the hole of triangle." |
| Pour water | M2 – Position: Free, Orientation: Axis | Color, Size, Relative Position | "Pour the water from the green mug to the red mug." |
| Wipe table | M2 – Position: Plane, Orientation: Axis | Color, Size, Directions, Relative Position | "Wipe the horizontal area with a sponge." |
| Use drawer | M2 – Position: Line, Orientation: Depend on position | Amount, Action Type, Level | "Fully close the top drawer." |
| Use door | M2 – Position: Fixed trajectory, Orientation: Depend on position | Amount, Action Type | "Slightly open the door." |

Table 2: The table contains the category-level tasks in our dataset, with their main constraints, variations and instruction samples.

**type** includes "open" and "close", especially for the articulated objects. The table of these variations and models used can be found in Appendix A.

**Unseen Settings** All tasks in the unseen settings are unseen <color, shape> combinations from an unseen color collection and an unseen shape collection (where the shapes include all object classes and variants). The unseen color collection has five new colors that do not appear during training, including brown, gold, pink, chocolate, and coral. As for the unseen shape collection, it has some overlap with the seen library for the tasks with color variations (but the <color, shape> combinations are always unseen), and is exclusive for the tasks without color variations (e.g., we introduce a new door model with a rotatable handle for the unseen setting of Door tasks). So there are mainly three kinds of unseen combinations: <unseen color,unseen shape>, <unseen shape>, and <unseen color,seen shape>. The exact object models used for each task can be found in Appendix A.

## 5  Vision-and-Language Manipulation Agent

To provide a baseline method to solve VLM tasks, we propose a neural-network-based agent 6D-CLIPort that takes in the input of multi-view RGB-D observations and task language descriptions and outputs 6 DoF pose keypoints along the path that can accomplish a specific task. For instance, for the pick-and-place task, the agent will iterate twice and output the object's 6D pose for pick, and place, separately.

The overall flow of 6D-CLIPort[2] is shown in Figure 4. The input data passes through visual and language encoders. The embedded features are fused together to obtain a pixel-wise feature map that shows the probability of the object of manipulation interest. The encoding module is reused in 3 models (Attention, Value, and Key Module) as follows: The Attention module takes the maximum of the feature map to get 2D image sample crops. The Key module takes the input of the rotated crops and output feature map crops. Then, the feature map crops are used as convolution kernels and pass through the feature map of the entire image from the Value module to get a 3D pose heatmap (x-y 2D position and yaw rotation, and their estimations are obtained by maximizing the probability). Three fully connected neural network modules will regress one of the remaining three dimensions each (z position and roll, pitch rotation) from this heatmap. Finally, a full 6 DoF pose is composed. Compared to the state-of-the-art method CLIPort [25], we resolved its two constraints: 1. increase 3 DoF only motions to full 6 DoF; 2. output an arbitrary number of poses than the fixed two outputs for pick and place actions. Besides, we introduce some details below.

---

[2]6D-CLIPort is implemented based on CLIPort [25] (Apache-2.0 License)

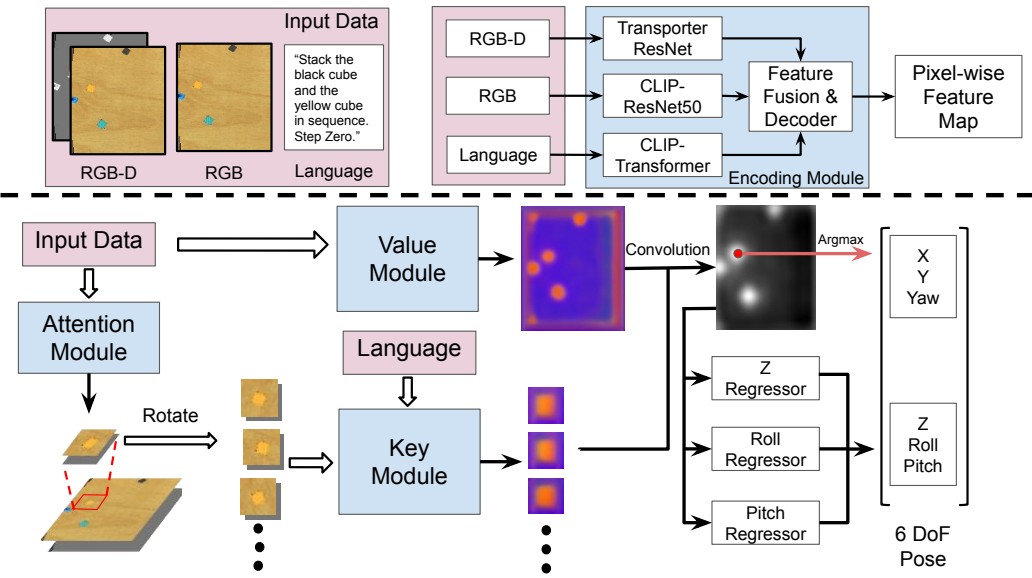

Figure 4: The structure of 6D-CLIPort. Please refer to Sec. 5 for details.

**Multi-view Vision Fusion** To get a better RGB-D input, we first fuse RGB-D input from several cameras with known poses into a 3D colored point cloud, and then project it along the vertical direction facing towards the table plane to get a top-view RGB-D image.

**Encoding Module** The agent has two feature extraction streams: semantic and spatial streams. The agent uses the pretrained CLIP's ResNet50 and Transformer model [22] to encode the RGB and languages in the semantic stream. An untrained Transporter ResNet [38], which has 43 layers and 8 strides, is used in the spatial stream to encode the RGB-D image. Then, the decoder fuses these latent space features by concatenation, fully convolution, and up-sampling layers and predicts a dense pixel-wise feature map. More details can be found in [25].

**Implementation Details** We separately train the agent for each task category. For example, the agent for pick and place tasks will jointly train on the data of all variations mentioned in Table 4.2. In details, since the VLMbench is built by the unit task templates, the input demonstrations $D = \{L, \zeta_1, \zeta_2, ..., \zeta_i\}$ can be divided into different sub demonstrations by the waypoints generated from unit task templates, where each step $\zeta_i = (L_i, o_i, g_i)$ consists of language instruction $L_i$, observation $o_i$, and the 6 DoF sub-goal waypoint pose $g_i$ for the current step. For step $i$, we use the observations in the first frame of this step as $o_i$ and prompt high-level instruction $L$ with "Step i" as $L_i$. The input RGB-D image has a resolution of $160 \times 128$. The feature heatmap is dimension 16 in *x-y* 2D plane, and the crops from the Attention Module are rotated 36 times before feeding into the Key Module.

# 6 Experiments

## 6.1 Experimental Setup

**Evaluation Settings** Before testing each baseline, we preprocess the tasks to help the agent eliminate trivial steps. 1) we divided tasks into the sub-goal sequences by the ground truth waypoints generated by AMSolver, so that the agent only needs to estimate the actions of the predefined unit task sequence for each task. 2) Since we have the ground truth gripper state for each sub-goal, we also provide whether the gripper should open or close for each action estimation. 3) To increase the grasping stability, we use the pre-generated grasping pose instead of estimations if these two poses are close enough (distance is less than 5 cm, and the rotation is less than 10 degrees). 4) To reduce the failure grasping cases due to the motion planning, we use a predefined pre-grasping offset, which is 8cm backward along the z-axis of the target grasping pose, and a post-grasping offset, which is 8cm upward along the z-axis of the world frame.

| Agent | Pick&Place | | Stack | | Drop | | Shape Sorter | |
|---|---|---|---|---|---|---|---|---|
| | Seen | Unseen | Seen | Unseen | Seen | Unseen | Seen | Unseen |
| Language-Only | 0.00 | 0.00 | 0.00 | 0.00 | 0.00 | 0.00 | 0.00 | 0.00 |
| Vision-Only | 6.31 | 9.85 | 6.89 | 1.79 | 0.00 | 0.00 | 0.00 | 0.33 |
| 6D-CLIPort | **28.28** | **27.53** | **22.19** | **18.37** | **6.42** | **6.42** | **17.33** | **12.33** |
| 6D-CLIPort (GT Ori) | 28.03 | 26.26 | 26.53 | 26.02 | 17.91 | 16.22 | 24.00 | 15.67 |
| 6D-CLIPort (GT Pos) | 83.84 | 75.25 | 58.93 | 50.51 | 16.89 | 11.82 | 18.00 | 17.33 |
| | Pour | | Wipe | | Door | | Drawer | |
| | Seen | Unseen | Seen | Unseen | Seen | Unseen | Seen | Unseen |
| Language-Only | 0.00 | 0.00 | 0.00 | 0.00 | 0.00 | 0.00 | 4.17 | 1.04 |
| Vision-Only | 0.00 | 0.00 | 19.80 | 20.80 | 0.00 | 0.00 | 14.58 | 7.29 |
| 6D-CLIPort | **1.00** | **1.00** | **22.40** | **21.00** | **6.00** | **5.00** | **22.92** | **15.63** |
| 6D-CLIPort (GT Ori) | 3.67 | 3.67 | 25.80 | 25.20 | 6.00 | 5.00 | 23.96 | 17.71 |
| 6D-CLIPort (GT Pos) | 0.33 | 0.67 | 60.20 | 53.40 | 27.00 | 27.00 | 43.75 | 52.08 |

Table 3: Success rates of all task categories, including both seen and unseen settings.

In each episode of one task variation, the simulator imports a test configuration from the pre-collected test dataset, which includes the initialization poses of objects, the success conditions of the task, and a set of waypoints that can finish the task for reference. The agent should solve the task in the online simulator within a limited number of steps. The success rate is used as the primary evaluation metric, calculated by dividing the number of success conditions satisfied by the number of tests. We use the average success rate of all variations for each task category. The success conditions are mainly determined by an object or joint detector. The object detector returns true if particular objects have moved inside the predefined space, and the joint detector returns true when the joint angle reaches the predefined range. The success conditions of each task can be found in Appendix A.

**Baselines** In addition to the 6D-CLIPort model, we provide two kinds of baseline models for comparison, one with partial input modalities and the other with partial ground-truth predictions.

To test the influence of different input modalities, we train two other agents with partial modalities: a *Language-Only* agent and a *Vision-Only* agent. These agents use the same model architecture as 6D-CLIPort but have different input modalities. The Language-Only agent uses the CLIP transformer for language encoding, and the visual inputs are all zeros. The Vision-Only agent uses the same RGBD inputs as in the 6D-CLIPort agent, but its language input will only include the prompt for steps indication like "Step zero" without any high-level language instructions.

To measure the capabilities and limitations of 6D-CLIPort, we individually test its position or orientation estimation abilities by giving the ground-truth values of the other. Although the trajectories of finishing the tasks are various, and we cannot obtain the optimal position and rotation information, we can regard the poses of waypoints as sub-optimal solutions. *GT Pos* means given ground truth *x/y/z* positions while the other three parameters for 3D orientation are estimated, and *GT Ori* suggests the contrary (known orientation, using estimated 3D position).

## 6.2 Result Analysis

**Main Results on Different Task Categories and Variations** The main results on different task categories are shown in Table 3. We observe that 6D-CLIPort performs better on the tasks that have lower rotation variances, including "Pick&Place," "Stack,""Shape Sorter," "Wipe," and "Drawer." It indicates that 6D-CLIPort can better estimate the positions than orientations in the 3D spaces. Moreover, 6D-CLIPort performs poorly on the "Pour" tasks since the task needs to adjust the pouring poses following the grasping pose, which introduces additional difficulties. Moreover, although the success rates in the unseen settings are generally lower than those in the seen settings, the performance drop is reasonable and not dramatic, showing that 6D-CLIPort can transfer the learned manipulation knowledge from seen objects to unseen objects, benefiting from the powerful transfer ability of the pre-trained CLIP model [22].

| Agent | Color | | Shape | | Size | | Relative Position | |
|---|---|---|---|---|---|---|---|---|
| | Seen | Unseen | Seen | Unseen | Seen | Unseen | Seen | Unseen |
| Language-Only | 0.00 | 0.00 | 0.00 | 0.00 | 0.00 | 0.00 | 0.00 | 0.00 |
| Vision-Only | 6.00 | 5.00 | 6.00 | 8.50 | 6.05 | 5.10 | 6.80 | 5.10 |
| 6D-CLIPort | **15.17** | **13.00** | **23.00** | **19.50** | **18.35** | **14.92** | **15.31** | **15.82** |
| 6D-CLIPort (GT Ori) | 18.67 | 17.67 | 28.00 | 27.25 | 20.97 | 17.74 | 21.43 | 18.37 |
| 6D-CLIPort (GT Pos) | 40.17 | 35.00 | 56.25 | 51.25 | 44.96 | 38.51 | 38.61 | 34.86 |
| | Direction | | Level | | Action Type | | Amount | |
| | Seen | Unseen | Seen | Unseen | Seen | Unseen | Seen | Unseen |
| Language-Only | 0.00 | 0.00 | 4.17 | 1.04 | 2.04 | 0.51 | 2.04 | 0.51 |
| Vision-Only | 21.00 | 24.00 | 14.58 | 7.29 | 7.14 | 3.57 | 7.14 | 3.57 |
| 6D-CLIPort | **22.00** | **26.00** | **22.92** | **15.63** | **14.29** | **10.20** | **14.29** | **10.20** |
| 6D-CLIPort (GT Ori) | 26.00 | 27.00 | 23.96 | 17.71 | 14.80 | 11.22 | 14.80 | 11.22 |
| 6D-CLIPort (GT Pos) | 53.00 | 41.00 | 43.75 | 52.08 | 35.20 | 39.29 | 35.20 | 39.29 |

Table 4: Success rates of all variations, including both seen and unseen settings.

We also show the results from the perspective of variations across task categories in Table 4. The results show that the agent is more sensitive to the novel compositions and thus has a larger seen-unseen performance drop on variations such as "Shape", "Level", "Action Type", and "Amount".

**Impact of Different Input Modalities** Table 3 also compares 6D-CLIPort with Language-Only and Vision-Only agents, demonstrating the impact of different input modalities. (1) The baseline vision-and-language manipulation model 6D-CLIPort performs the best on all tasks, showing the importance of both visual observations and language guidance in VLMbench tasks. (2) The Language-Only agent nearly fails at all the tasks as it is visually blind and thus unable to localize the objects in the 3D space. For "Drawer" tasks, since language instructions can provide the level information (e.g., top, middle, and bottom) and action directions (e.g., open and close), the Language-Only agent has some chance to close the drawer by collisions. (3) Without language guidance, the Vision-Only agent has a significant performance degradation on all tasks. It fails completely on tasks that require more strict pose constraints, including "Drop", "Shape Sorter", "Pour", and "Door". For other tasks such as "Pick&Place", "Stack," "Wipe," and "Drawer", the Vision-Only agent can succeed in few cases (though with pretty low success rates) by randomly grasping an object in the scene for manipulation because those tasks have a lower variance of the grasping actions and following movements.

**Ablation Study on Position and Orientation Estimation** We provide partial ground-truth predictions and do a unit test of the agent's position and orientation estimation abilities. The results are shown in Table 3 and Table 4. From the results, we can see the position estimation ability significantly limits the performance of 6D-CLIPort in those tasks which need correct object localization, such as "Pick & Place", "Stack", and "Shape Sorter" tasks. One primary reason why the GT position brings more improvement is that it eliminates the difficulties of localizing the target object, one of the main challenges in compositional reasoning. For example, the instruction "place the red cube into the green container" requires the model to localize the correct cube and container and providing GT position makes the task much easier. Besides, from the perspective of pose variances, when we divide the task into steps, the orientation of each step has fewer variances than the position in many tasks, such as picking, stacking, and wiping. Furthermore, for the tasks requiring the cooperation of position and orientation, such as "Pour" and "Door" tasks, we observe that giving partially ground truth may still not guarantee better task completion results. More results and analysis can be found in Appendix D.

## 7 Conclusion and Future Work

Vision-and-Language Manipulation (VLM) tasks are essential since they are inevitable for embodied AI. For further research in this area, we propose VLMbench, which includes various VLM tasks, and AMSolver, used for automatic VLM task generation. In addition, we test the 6D-CLIPort agent, a keypoint-based 6 DoF agent, on the benchmark. The results show that the current models can finish VLM tasks, but it is still a new area needed to be explored. We hope the VLMbench can push the research to find general language-guided manipulation agents.

**Limitations** Our work still has limitations that can be improved by future work. First, we only consider rigid body object manipulation in the VLMbench. It is important to include soft material objects in the future. Second, indirect manipulation tasks, like throwing the ball and playing billiards, are not included in the VLMbench. Third, since we generated data with template languages in the simulator, the gap between the virtual environment and the real world cannot be ignored.

**Ethical Concerns** We do not see significant risks of security threats or human rights violations in our work. Since our work contributes to the field of language-guided manipulations, we do not encourage real-world robot experiments depending on our benchmark without any real-world data fine-tuning. Due to the gap between the simulator and the real world, the agents may execute unexpected actions.

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
