# OpenReview forum: "VLMbench: A Compositional Benchmark for Vision-and-Language Manipulation"
_NeurIPS.cc/2022/Track/Datasets_and_Benchmarks — NeurIPS 2022 Datasets and Benchmarks _

### Official Review · Reviewer_5nmE · 2022-07-07
**Benchmark and dataset for vision and language manipulation**

**Rating:** 7
**Confidence:** 4

**Strengths:**

I believe the authors did a good job in formalising various manipulation tasks. The work proposes a systematic way to describe new tasks in robotic manipulation problems (Sec. 3.1). I think the modular design suggested in the paper is a good idea. Object-centric representations along with unit task builder allow for the scalability of data generation.

AMSolver seems to provide a flexible way to generate demonstrations for new tasks after having defined proper goal constraints. I believe providing automatic 6DoF grasping is a big benefit of the tool.

The dataset provided with the benchmark offers a good variety of manipulation scenarios.

A baseline and ablation suggested by the authors are sufficient to emphasise the challenges of the proposed benchmark.

**Weaknesses:**

Given the proposition of the dataset, I would expect more details on the collected data to be provided. Some of the details are covered in the appendix. However, it would be good to discuss what set of objects is used in the tasks (list of objects, do they come from the existing dataset) and how many different objects can be used per task (e.g. how many can be opened).

Some more details on how the initial scenes were randomised would be also useful (how many objects, were placed randomly/with constraints). Additionally, what is the process of creating the instructions from templates? Is it similar to CLEVR [1]?

Additionally, for the dataset description, what is the distribution of demonstrations over the tasks?

There are 5 different views generated in the tool, and a description of what views are incorporated, and why such views were chosen, would be a nice addition. Also, what is the exact set of observations provided in the system output?

For the benchmarking, it would be good to discuss the performance with respect to FPS that can be achieved. Parsing 5 images per 50ms may provide a noticeable delay.

Sec. 6.1. suggests that multi-step tasks are provided with ground truth sub-goals. I think it would be worth showing the model dealing with multi-step tasks, and then maybe showing how compositioning of simple tasks affects the performance, and probably provides a significant challenge to the proposed model.

I believe that the issues I mentioned are of a rather minor nature, and are easily correctable.

[1] Justin Johnson, Li Fei-Fei, Bharath Hariharan, C Lawrence Zitnick, Laurens Van Der Maaten, and Ross Girshick. CLEVR: A Diagnostic Dataset for Compositional Language and Elementary Visual Reasoning. In IEEE Conference on Computer Vision and Pattern Recognition (CVPR), 2017.

**Additional Feedback:**

Questions:
- How is the grasp pose chosen in a particular scenario? Are grasp randomly chosen from available ones, or is there an optimisation w.r.t. grasp choice?
- How are [target object description]s generated? How are the variations of the descriptors chosen (e.g. is it always all of *colour, size, shape*, or a random subset is used)?

**Clarity:**

Overall, the paper is written clearly. I believe some details mentioned in the Weaknesses section, could be moved to the main article from the appendix, and some details could be explained a bit more.

**Correctness:**

The dataset is constructed in a sound way.  The 8 manipulation tasks chosen, are of different nature and provide various challenges for robotic systems.

**Documentation:**

I believe the documentation is good enough for the use and reproducibility of the dataset.

**Ethics:**

No ethical concerns.

**Relation To Prior Work:**

Relation to prior work is well covered.

**Summary And Contributions:**

The authors of VLMbench propose a new benchmark for robotic manipulation based on visual input and a language query. A benchmark assesses the completion of the task based on task-related final state constraints. Additionally, the work proposes a new dataset associated with the benchmark. The authors propose 8 various manipulation tasks. For each task, a set of randomised
template-based scenarios is defined. Additionally, the paper introduces AMSolver which is an automatic tool for demonstration generation.

---

> ### Author Response · Authors · 2022-08-17
> **Responses to reviewer 5nmE**
>
> 1. > what set of objects is used in the tasks (list of objects, do they come from the existing dataset) and how many different objects can be used per task (e.g. how many can be opened).
>
>     The set of objects can be divided into five types and 22 classes, which are listed below. The models used in VLMbench mainly come from ShapeNet [1]. The number behind the object class indicates the number of instances inside that class. The objects used per task have been added to Appendix A in the revision.
>
>     | Object types | Number of classes | Classes |
>     | :-----| ----: | :----: |
>     | Basic model | 3 | cube (1), triangular prism (1), cylinder (1) |
>     | Special model | 9 | star (1), moon (1), cross (1), flower (1), letter of 't' (1), pencil (1), basket (1), box container(1), shape sorter (1) |
>     | Planar model | 6 | rectangle (1), circle (1), triangle (1), star (1), cross (1), flower (1) |
>     | Functional model | 2 | mug (6), sponge (1) |
>     | Articulated model | 2 | door with one rotatable handle (2), cabinet with three vertical drawers (3) |
>
> 2. > Some more details on how the initial scenes were randomised would be also useful (how many objects, were placed randomly/with constraints).
>
>     The object numbers vary among the task categories and variations. In general, we have one target object and at least one distractor in the scene. At the beginning of tasks, the colors, scales, and positions of all objects will be randomly set. We have added the details of the initial scenes for each task variation in Appendix A of the revision.
>
> 3. > Additionally, what is the process of creating the instructions from templates? Is it similar to CLEVR [2]?
>
> 	Yes. The instruction generation process is similar to the CLEVR [2]. The language templates can automatically acquire the attributions of objects and fill them into the corresponding positions.
>
> 4. > what is the distribution of demonstrations over the tasks?
>
>     We list the number of episodes for each task category below. For more details on each variation, please refer to Table 7 in the revision.
>
>     | Task category | Train | Validation (seen and unseen) | Test (seen and unseen)|
>     | :--:| :--: | :--: | :--: |
>     | Pick | 900 | 370 | 792|
>     | Stack | 940 | 390 | 784 |
>     | Drop | 795 | 325 | 592 |
>     | Place | 560 | 210 | 600 |
>     | Wipe | 640 | 235 | 1000|
>     | Pour | 508 | 185 | 600|
>     | Door | 200 | 40| 200|
>     | Drawer | 240 | 120| 192|
>     | Total | 4783 | 1875 |4760|
>
> 5. > why such views were chosen, would be a nice addition
>
>     We refer to the camera settings in RLBench [3]. Four stereo cameras from the overhead, front, left shoulder, and right shoulder can provide full environment observation. One eye-in-hand camera can provide the information for the partial observation research.
>
> (Continue to the next comment)

---

> > ### Author Response · Authors · 2022-08-17
> > **Responses to reviewer 5nmE (continue)**
> >
> > 6. > Also, what is the exact set of observations provided in the system output?
> >
> >     The observation of each step contains RGB-D images from five views, object poses, end-effector poses, joint angles, velocities, and forces. The user may get the point clouds and instance segmentations from five camera views if needed.
> >
> > 7. > FPS that can be achieved
> >
> >     If the simulator needs to render at every step, it is close to 10 FPS with GPU acceleration on one process.
> >
> > 8. >  I think it would be worth showing the model dealing with multi-step tasks, and then maybe showing how compositioning of simple tasks affects the performance, and probably provides a significant challenge to the proposed model.
> >
> >     Thanks for the good advice. In Table 6 of the original submission and Tables 8 & 9 of the revision, we have shown the success rates of each step for every agent. For tasks in the VLMbench, each task can be considered as a multi-step task since each task at least consists of one grasping step and one following movement step. Therefore, we used two metrics for goal-conditioned per step success: Goal-condition Grasp success rate and Goal-conditioned Movement success rate. The Goal-conditioned Grasp (G-G) success rates indicate whether the agent has grasped the correct object in the first step. The Goal-conditioned Movement (G-M) success rates indicate whether the agent can satisfy the success conditions in the following movement by using the pre-generated grasp poses for the grasping step. More analysis can be found in Appendix D of the revision.
> >
> > 9. > How is the grasp pose chosen in a particular scenario?  Are grasp randomly chosen from available ones, or is there an optimisation w.r.t. grasp choice?
> >
> >     The grasp pose of one demonstration starts by sampling randomly from the available ones. Then, we use the motion planning library, OMPL, to generate the following trajectories. If the planner cannot find a feasible path until reaching the attempt limitations, the next grasp pose will be tried.
> >
> > 10. > How are [target object description]s generated? How are the variations of the descriptors chosen (e.g. is it always all of colour, size, shape, or a random subset is used)?
> >
> >     The target object descriptions are generated by predefined rules, and we only evaluate one variation in an instance-level task. For example, in the ‘Pick&Place: Color’ tasks, two same-shape models and two same-shape containers are in the scene initialization. All colors with the name and RGB values are randomly sampled from the color library. Then, the object description is "[color1/color2] object" and the container description is "[color3/color4] container".
> >
> > References:
> >
> > [1] Chang, Angel X., et al. "Shapenet: An information-rich 3d model repository." arXiv preprint arXiv:1512.03012 (2015).
> > [2] Johnson, Justin, et al. "Clevr: A diagnostic dataset for compositional language and elementary visual reasoning." Proceedings of the IEEE conference on computer vision and pattern recognition. 2017.
> > [3] James, Stephen, et al. "Rlbench: The robot learning benchmark & learning environment." IEEE Robotics and Automation Letters 5.2 (2020): 3019-3026.
> > [4] Sucan, Ioan A., Mark Moll, and Lydia E. Kavraki. "The open motion planning library." IEEE Robotics & Automation Magazine 19.4 (2012): 72-82.

---

### Official Review · Reviewer_3Bt1 · 2022-07-22
**I cannot reproduce the authors' results, and I am conservative in accepting this paper**

**Rating:** 7
**Confidence:** 3

**Strengths:**

1. I think the dataset is important for the language and robotics/vision research community and makes a step towards language-guided 6DoF robot manipulation. Existing 6DoF datasets, such as [ManipulaTHOR](https://github.com/allenai/manipulathor), [CALVIN](https://github.com/mees/calvin), etc., usually focus on long horizontal task planning. While [Ravens](https://github.com/google-research/ravens) is suitable for language-guided manipulation, it only operates on a 2D plane. Therefore, I think VLMBech fills the gap of language-guided 6DoF robot manipulation well.

2. The authors propose and build a complete pipeline for language-guided 6DoF robot manipulation research, including a simulator  AMSolver, a dataset VLMBench, and a baseline model 6D-CLIPort, which can help novices get to know this field quickly and researchers develop and compare algorithms conveniently.


**Weaknesses:**

1. Following the GitHub repository of this paper, it is hard to reproduce the results (Table 3 and Table 6 in the appendix). For instance, I used the pre-trained model for the 'stack' task, and always got a 0 success rate.

2. The process of generating the dataset is not described very clearly. How many episodes for each task in training? What is the difference between seen and unseen? How is the training and validation split? Following the GitHub repository of this paper, I think that running 'python tools/dataset_generator_NLP.py' only gives the validation dataset?

3. Some parts of the paper are not written very well. For the AMSolver, how do the three modular rule-based constraints correspond to the three general household task categories?
Some literature on language-guided robot manipulation, such as [1], is missing. The conclusion is missing.

[1] Mees et al. What Matters in Language Conditioned Imitation Learning. IROS 2022.

**Additional Feedback:**

In my opinion, 6DOF robot manipulation and long horizontal task planning are very different. Adding long horizontal planning may introduce additional complexity to the task. The authors may consider it when designing tasks.

I like the paper and appreciate the efforts made by the authors. Please add the missing details and proofread the paper more carefully. Thanks.

**Clarity:**

The paper is generally easy to follow and well written. But for the AMSolver, the correspondence of the three modular rule-based constraints and the three general household task categories are not written very clearly. Some literature on language-guided robot manipulation, such as [1], is missing. The conclusion is missing.

Some typos:

1. Line 25, Additional -> Additionally

2. Line 26, the biases -> biases

3. Line 60, DOF -> DoF. Please check other DOF or DoF, both are fine, but please be consistent.

4. Line 67, demonstration -> demonstrations

5. Line 98, object -> objects

6. Line 100, considered -> considered as

7. Line 141, with -> while

8. Line 171, receive -> receives, generate -> generates

9. Line 179, tasks -> task

10. Line 185, Tasks -> Task

11. Line 191, the format of double quotes not right

12. Line 217, entire image -> the entire image

13. Line 231, extra space after the word concatenation

14. Line 245, we -> We

15. Line 284, given -> giving


[1] Mees et al. What Matters in Language Conditioned Imitation Learning. IROS 2022.


**Correctness:**

The authors first build a simulator AMSolver and use it to construct the dataset VLMBech. The AMSolver adopts modular rule-based task templates to classify and construct tasks, including control constraints, moving the target object with goal pose constraints (M1), and moving the target object along a trajectory while satisfying the motion constraints (M2). The design of variations adopts an object-centric representation, including class, color, size, and geometry shape. The rationale for constructing the dataset is sound. However, the details of how to generate the dataset, including the number of episodes, and the training and validation split, are missing.

The paper also proposed a baseline 6D-CLIPort and conducted experiments on it. The experiment design is appropriate. However, following the Github repository, it is hard to reproduce the main results (Table 3 and Table 6 in the appendix), and the paper also misses experimental details for reproducing the results. Additionally, for Tables 3 and 4, what does the unimodal mean?


**Documentation:**

I think the authors did not provide sufficient detail on data collection and organization, including the number of episodes for each task, the difference between seen and unseen, the training and validation split, and so on. The dataset in Google drive is different from the dataset generated by the script ‘python tools/dataset_generator_NLP.py’. The paper also misses experimental details for reproducing the results. Additionally, following the Github repository, I still cannot reproduce the main results (Table 3 and Table 6 in the appendix). Therefore, I am conservative in accepting this paper.

**Ethics:**

I do not see obvious risks of security threats, human rights violations, or any other potential ethical problems in this work.

**Relation To Prior Work:**

In general, the paper clearly discusses the difference between this work and previous contributions. The literature [1] should be added. In addition, the authors may consider distinguishing between 6DOF robot manipulation and long horizontal task planning, which in my opinion is very different, and long horizontal task planning may introduce additional complexity to the task.


[1] Mees et al. What Matters in Language Conditioned Imitation Learning. IROS 2022.

**Summary And Contributions:**

This paper considers the problem of 6 degrees of freedom (DoF) robot manipulation following language instructions, which is an interesting and promising direction, and there are several works emerged recently. In particular, the authors proposed a simulator AMSolver based on the robot learning framework RLBench. The authors added the language instruction part in RLBench and proposed modular rule-based task templates to classify and construct tasks, including control constraints, moving the target object with goal pose constraints (M1), and moving the target object along a trajectory while satisfying the motion constraints (M2). An object-centric representation is used to add variations when generating data, including class, color, size, and geometry shape. Based on AMSolver, the authors constructed the dataset VLMBench for 6DoF language-guided robot manipulation research. Based on CLIPort, the authors further proposed a 6DoF keypoint-based model as a baseline for VLMBench.

In summary, this paper considers an important problem, language-guided robot manipulation, and the contributions consist of a simulator AMSolver, a dataset VLMBench, and a baseline model 6D-CLIPort.

---

> ### Author Response · Authors · 2022-08-17
> **Responses to reviewer 3Bt1**
>
> We thank the reviewer for appreciating the importance and contributions of this work. We have incorporated the suggestions regarding the citation, conclusion, and typos in the revision. Below we hope to address other questions in the review. Feel free to let us know if you have any further questions!
>
> 1. Re: Result reproduction
>
>     To fix a potential bug, we updated a more balanced dataset as well as the new model weights on the GitHub repository. We also updated the new experimental results in the revision (which consistently support the claims in the paper). To fully ensure that the results are reproducible, we asked several third-party researchers to follow the instructions to run the model, and all of them managed to reproduce the results. We hope the reviewer can successfully run the model now, and feel free to let us know if there are still some difficulties reproducing the results. We would be very glad to provide any additional guidance.
>
> 2. Re: Dataset generation
>     > How many episodes for each task in training?
>
>     We list the number of episodes for each task category below. For more details on each task variation, please refer to Table 7 in the revision.
>
>     | Task category | Number of episodes |
>     | :--:| :--: |
>     | Pick | 900 |
>     | Stack | 940 |
>     | Drop | 795 |
>     | Place | 560 |
>     | Wipe | 640 |
>     | Pour | 508 |
>     | Door | 200 |
>     | Drawer | 240 |
>     | Total | 4783 |
>
>     >  What is the difference between seen and unseen?
>
>     The explanations of unseen settings can be found in Appendix D of the original submission and Section 4.2 in the revision. All tasks in the unseen settings are unseen <color, shape> combinations from an unseen color collection and an unseen shape collection (where the shapes include all object classes and variants).
>     The unseen color collection has five new colors that do not appear during training, including brown, gold, pink, chocolate, and coral.
>     As for the unseen shape collection, it has some overlap with the seen library for the tasks with color variations (but the <color, shape> combinations are always unseen), and is exclusive for the tasks without color variations (e.g., we introduce a new door model with a rotatable handle for the unseen setting of Door tasks). So there are mainly three kinds of unseen combinations: <unseen color,unseen shape>, <unseen shape>, and <unseen color,seen shape>.
>     The exact object models used for each task can be found in Appendix A in the revision.
>
>     > How is the training and validation split?
>
>     The dataset statistics can be found in Appendix C. The benchmark has 6,555 manipulation demonstrations, consisting of 4,680 training demonstrations, 1,170 seen validation demonstrations, and 705 unseen validation demonstrations. The training and seen validation sets are sampled from the same task configurations so that the episodes are randomly split. Meanwhile, the unseen validation set shares the same task configuration as the unseen test set.
>
>     > Following the GitHub repository of this paper, I think that running 'python tools/dataset_generator_NLP.py' only gives the validation dataset?
>
>     You can change the configurations on the top of  ‘tools/dataset_generator_NLP.py' to switch different split generations. The color and object collections in the ‘amsolver/const.py’ can be used to generate the seen and unseen settings. We have updated the comments in the code.
>
> 3. > For the AMSolver, how do the three modular rule-based constraints correspond to the three general household task categories?
>
>     To explain the connections between unit tasks and general household tasks, we first want to clarify that AMSolver can be considered as a constructive system to represent a set of manipulation tasks. In general, the manipulation tasks mainly need to consider two parts: the constraints and objects. In other words, the manipulation tasks are to move the target objects with certain constraints. Benefiting from the constraints definition of AMSolver, we can decompose a general household task into unit tasks belonging to the templates, where the connection of different unit tasks defines the constraints automatically. For instance, the example shown in Figure 1 ‘take out the plate from the dishwasher’ can be divided into several steps, and each step can be represented by a unit task. We have reorganized the AMSolver section in the revision to make this clear.
>
> (Continue to the next comment)

---

> > ### Author Response · Authors · 2022-08-17
> > **Responses to reviewer 3Bt1 (continue)**
> >
> > 4. Re: Distinguishing between 6DOF robot manipulation and long horizontal task planning
> >
> >     From the perspective of challenges, long-horizon task planning mainly focuses on how to plan the sub-task sequence. For example, long-horizon task planning aims to divide "put the cube into the drawer" into "open drawer," "pick up the cube," and "place the cube in the drawer." Each sub-task may only have very few variations [2] or even be assumed to finish automatically [3]. In contrast, 6 DoF robot manipulation aims to generate executable manipulation trajectories for the robot arm to finish the task according to the object properties and constraints. For example, a 6 DoF robot manipulation agent should be able to open the door regardless of the shape, pose correctly, or properties. We agree with the reviewer that adding long horizontal planning will introduce additional complexity to the task. In fact, each sub-task in the long horizontal planning tasks can be considered as an individual manipulation task, and the performance of these sub-tasks will directly influence the success of the whole long-term task. Therefore, in reality, 6 DoF robot manipulation tasks are the prerequisites for executing the long-horizon tasks. Meanwhile, long-horizon tasks introduce the causal relationship between individual manipulation tasks.
> >
> > 5. > Additionally, for Tables 3 and 4, what does the unimodal mean?
> >
> >     The explanations of other agents can be found in Section 6.2 in the revision. In order to test the influence of different input modalities, we also trained two unimodal agents: a Language-Only agent and a Vision-Only agent. These agents use the same structure as 6D-CLIPort but have different input modalities. The Language-Only agent uses the CLIP transformer for language encoding, and the image inputs are all zeros. And the Vision-Only agent uses the same RGBD inputs as in the 6D-CLIPort agent, but the language will only include the prompt for step indication without high-level language instructions, like ``Step zero".
> >
> > References:
> >
> > [1] Mees, Oier et al. “What Matters in Language Conditioned Robotic Imitation Learning.” ArXiv abs/2204.06252 (2022): n. Pag.
> > [2] Mees, Oier, et al. "CALVIN: A Benchmark for Language-Conditioned Policy Learning for Long-Horizon Robot Manipulation Tasks." IEEE Robotics and Automation Letters (2022).
> > [3] Shridhar, Mohit, et al. "Alfred: A benchmark for interpreting grounded instructions for everyday tasks." Proceedings of the IEEE/CVF conference on computer vision and pattern recognition. 2020.

---

> > > ### Comment · Reviewer_3Bt1 · 2022-08-28
> > > **Thanks for your elaborate responses**
> > >
> > > Many thanks for your elaborate responses. I am now more confident about this work and have updated my score accordingly. But I still have some minor questions regarding to data generation as follows:
> > >
> > > 1. If I run `python tools/dataset_generator_NLP.py`, will I get the same dataset as in https://drive.google.com/drive/folders/1Qx_2_ePIqf_Z6SnpPkocUiPgFeCfePQh? It seems the seeds are random. I suggest to save the seeds so that people do not need to download the full dataset (which is big), and generate it themselves.
> > >
> > > 2. What configurations do I need to change to get train, valid, and test datasets, respectively? Only the path?
> > >
> > > Sorry that I cannot access to my server recently to run the model. But I think it may be my mistake to check the performance table.
> > >
> > > Again, thanks for the work and responses.

---

> > > > ### Author Response · Authors · 2022-08-28
> > > > **Further responses to reviewer 3Bt1**
> > > >
> > > > Thank you for your time and your appreciation of this work!
> > > >
> > > > 1. > "It seems the seeds are random. I suggest to save the seeds so that people do not need to download the full dataset (which is big), and generate it themselves."
> > > >
> > > >    Thanks for your suggestion! Currently, the github repo supports dataset generation but may not guarantee that the generated data is exactly the same. But we will add more scripts to support this function. Meanwhile, we will put the dataset in different venues (e.g., Google Drive, AWS Storage, etc.) for easier access and download. We will also write a script to support faster and more stable downloads.
> > > >
> > > > 2. > "What configurations do I need to change to get train, valid, and test datasets, respectively? Only the path?"
> > > >
> > > >     For train and seen validation datasets, the changes include the path and episode number. For the unseen validation dataset, the configurations in the 'amsolver/const.py' also need to be changed. For the test dataset, since the episodes in the test dataset do not need to render the full demonstrations, there is another script used to generate the test dataset: 'tools/test_config_generator.py'. The usage is the same as 'tool/dataset_generator_NLP.py'.

---

> > > ### Comment · Reviewer_3Bt1 · 2022-08-28
> > > **Thanks for your elaborate responses**
> > >
> > > Many thanks for your elaborate responses. I am now more confident about this work and have updated my score accordingly. But I still have some minor questions regarding to data generation as follows:
> > >
> > > 1. If I run `python tools/dataset_generator_NLP.py`, will I get the same dataset as in https://drive.google.com/drive/folders/1Qx_2_ePIqf_Z6SnpPkocUiPgFeCfePQh? It seems the seeds are random. I suggest to save the seeds so that people do not need to download the full dataset (which is big), and generate it themselves.
> > >
> > > 2. What configurations do I need to change to get train, valid, and test datasets, respectively? Only the path?
> > >
> > > Sorry that I cannot access to my server recently to run the model. But I think it may be my mistake to check the performance table.
> > >
> > > Again, thanks for the work and responses.

---

### Official Review · Reviewer_gZvZ · 2022-07-24
**The presented paper provides a needed benchmark for evaluating embodied agents that should perform various manipulation tasks based on the language instructions. The tasks are restricted to the tasks where an object is manipulated.**

**Rating:** 8
**Confidence:** 4

**Strengths:**

I fully agree with the authors that such benchmark is filling an important blank spot to measure quality of embodied agents who should follow language instructions to perform object manipulation tasks. While there are several benchmarks and datasets for language descriptions to images, there are not that many for the embodied domain, especially not those that would enable measuring the quality in an organized structured way.
The paper is well organised and describes clearly the presented benchmark. The benchmark itself is from my perspective very well designed. It enables evaluation of the quality of individual subtasks and more complex tasks can be created by joining the individual subtasks.
I appreciate that the evaluation is enabled by the design of the dataset. All the tasks are specified by the change in position and orientation and therefore the ground truth waypoints are known. The evaluation is done by checking if individual waypoints have been reached.
The installation guide, datasheet file as well as appendix information are very helpful and detailed.


**Weaknesses:**

It is claimed in the paper that more complex tasks can be built as a composition of the simple tasks. However from the text itself it is not clear how they are built and evaluated.
The actions are restricted to the manipulation with an object - start of the action is determined by grasping an object and the end by releasing. Therfore actions such as push/kick/pull/hammer/etc. are not included. It is also not clear how these could be included (is there a way how to add a new task?) and evaluated. I would appreciate if the authors could discuss on this.
I am also not sure how the difficulty of the tasks might be compared or evaluated. Do the authors consider that all the tasks are on the same level? Is there any way to evaluate the difficulty of the individual tasks? (e.g., number of the waypoints needed to pass?)
The dataset itself is quite big to download (aprox. 10 GB). It might be good to include also smaller version for users who want to just test it and see samples from the dataset.

I was able to run the pretrained models, generate new data and test configurations as described in the readme. However, it is not clear how to turn on the simulator - is there any argument to do so? The idea/process is described well in the paper, but it is hard to navigate in the code as there is not any detailed documentation. I would appreciate more detailed documentation for the code itself.

**Additional Feedback:**

I really appreciate the creation of such a benchmark. I would like to see if the benchmark is extended for more tasks and if the authors provide a comparison of current algorithms/methods on the website as people will start to use the benchmark. Further questions/concerns are mentioned in the sections above.

**Clarity:**

The paper is very clearly written, easy to follow and well structured.
There are some grammar mistakes and typos (e.g., wiper table instead of wipe table (in Table 2); M2[a4,c2] on line 142 - I think should be M2[a4,b2];the agent receive -> receives (line 171); "consist of language instruction" -> "consists of language instruction"; "shape Sorter," -> "Shape Sorter", (line 273); "we can find out the position" -> "...that the position" on line 287; rotations(e.g....) -> rotations (...) - missing space (line 288))
- x, y, z - should be in italics font $x$...


**Correctness:**

The described methods are sound, the dataset is designed well. The benchmark evaluation is clear. I think that as a benchmark, the authors should present on the webpage or github the baseline and provide a table of the results of other methods (encourage their upload, etc.). Maybe in the future the authors might consider if it would be possible to provide an automated evaluation of any uploaded method.

One question considering the described methods is that I am not sure how the object detector is working. It is mentioned that the object location in the end is evaluated, but I am not sure about the details. Maybe this could be also described in the appendix.


**Documentation:**

The documentation is very clear. Both the dataset datasheet, appendix, and the GitHub webpage provide clear information about the presented benchmark with a sufficient amount of detail on most levels (some missing remarks are summarized in the "Weaknesses" section). All the necessary information about maintaining, hosting, and licensing is included.



**Ethics:**

I don't see any ethical concerns.

**Relation To Prior Work:**

The presented paper is very well put in the context with the prior work.


**Summary And Contributions:**

The presented paper proposes a robotics manipulation benchmark containing language instructions and corresponding manipulation tasks (e.g. pick and place, wipe a table, open a drawer, etc.).
The main contributions of the paper are:
1) Automatic generator of individual demonstrations with corresponding linguistic instructions. The demonstrations might be composed of simple tasks. There are 8 basic task types (pick & place objects, stack objects, drop pencil, put into shape sorter, pour water, wipe table, open drawer, use door) that can be varied by color, shapes, size, relative positions, etc.
2) VLMbench - benchmark including a set of manipulation tasks that compose of visual observations accompanied by linguistic instructions. The manipulation tasks have several variations and can be automatically evaluated.
3) Baseline algorithm evaluated on VLM tasks.

---

> ### Author Response · Authors · 2022-08-17
> **Responses to reviewer gZvZ**
>
> We thank the reviewer for appreciating the motivation, presentation, significance, and contributions of this work. We have incorporated the suggestions regarding the presentation and typos in the revision. Below we hope to address other questions in the review. Feel free to let us know if you have any further questions!
>
> 1. Re: Building complex tasks from simple tasks
>
>     To create complex tasks, like multi-step tasks, the user can compose a task template from the unit library of formulation, such as putting the object into the drawer can be formatted as Control-M2-Control-M1, and then select the objects from a given set to be included in the scene as the objects to be manipulated and the distractor objects. To evaluate the tasks, the success conditions of the tasks need to be predefined. For example, we can add an object detector in the drawer to evaluate whether the target object is put inside the drawer finally.
>
> 2. Re: Adding new tasks
>
>     As mentioned in Section 3.1, the 'control' step can be extended to other ways, like pushing and pulling. To generate the ground truth of these actions, we need to consider both the pushable and pullable points according to object geometries. Then, the following steps can be considered as object movement tasks with path constraints in a 2D plane. For dynamic tasks like kicking, they should consider both object dynamics and geometries. To make the object reach the target position,  we should calculate the initial position, orientations, velocity, and force by the object's dynamic properties. All these tasks can be evaluated by checking the object's position about whether the object maintains the movement on the ground. For hammering, it still needs to grasp the handle and move the impact surface of the hammer along the direction of the target force, so we can think it can be considered as an M2[a3, b4] task. This task can be evaluated by adding the force sensor on the target object and hammer surface to check whether the hammer correctly hit the object.
>
> 3. Re: The difficulties of tasks
>
>     No, we do not think all tasks are on the same level.
>     - First, we can compare the task difficulties from the perspective of constraints: (1) The tasks with path constraints should be more challenging than others. For example, wiping the table should be more complicated than the pick/place task since the agent must maintain the object close to the table and may follow a specific path as requested during the process. Waypoint checking will be a good idea to evaluate if it follows the desired path in some cases. (2) The tasks with strict position and orientation constraints should be more challenging than others. For example, stacking the cubes requires the object to be in stable contact with others instead of just being placed in a box without any constraints.
>     - Second, task difficulties can also be evaluated from compositional language reasoning. For example, distinguishing a red cube from a pink cube is more challenging than from a yellow star.
>     - Moreover, the tasks with more steps should be more challenging than others. For example, opening the door requires rotating the handle to unlock the door first, which increases the manipulation reasoning difficulties compared to opening the drawer.
>     - Another metric that might help evaluate the difficulties is the feasible space of the task completion trajectory. We think one task is more demanding than another if the feasible task space is smaller and narrower.
>
> 4. Re: Smaller samples of the dataset
>
>     Thanks for the suggestion. We have uploaded a smaller dataset version, which contains one episode for each task variation. The link can be found on GitHub.
>
> (Continue to the next comment)

---

> > ### Author Response · Authors · 2022-08-17
> > **Responses to reviewer gZvZ (continue)**
> >
> > 5. > However, it is not clear how to turn on the simulator - is there any argument to do so?
> >
> >     For visualization, you can set ‘headless=True’ when you initialize the simulator. We have added comments in the ‘cliport_test.py”.
> >
> > 6. >  I would appreciate more detailed documentation for the code itself.
> >
> >     Thanks for the suggestion. You can find the code about unit tasks in the ‘amsolver/backend/unit_task.py’ file, and the task templates are saved in the ‘vlm/tasks’ folder. We will add more documentation and code comments to the GitHub repository.
> >
> > 7. > Maybe in the future the authors might consider if it would be possible to provide an automated evaluation of any uploaded method.
> >
> >     Great idea. We are considering updating our project website and adding the evaluation interface in the future. We will post the news on the website and GitHub.
> >
> > 8. > One question considering the described methods is that I am not sure how the object detector is working.
> >
> > 	The object detector is a 3D box where the poses (positions and orientations) and properties (length, width, height) can be modified. The object detector is placed in the target destinations of the task. When the target object mesh is overlapped with the detector, it will return the true value for checking the object. We have added more details about this in Appendix B in the revision.

---

> > > ### Comment · Reviewer_gZvZ · 2022-09-02
> > > **Still believe that the paper is of a good quality and ready to be published**
> > >
> > > I would like to thank authors for clarification of my concerns. After extensively testing the VLMbench and reading the answers of the authors, I still believe that the work (including the code and documentation) is of good quality and ready to be published.

---

### Official Review · Reviewer_DdfD · 2022-07-28
**Review of VLMBench**

**Rating:** 6
**Confidence:** 3

**Strengths:**

- The paper presents a benchmark for a domain that is of significant interest to the community.
- The simulator/dataset allows for specifying constraints of several kinds - such as position/orientation constraints, constraints related to the trajectory executed by the agent.
- Similarly, the tasks have a good amount of variations possible in sizes, shapes, directions of motion etc. Linking such high fidelity task specification with language has the potential to enable more complex architectures and solution methodologies for vision and language manipulation.

**Weaknesses:**

- The paper identifies the compositional nature of task specification as a key highlighting feature of the benchmark, but it is unclear how this is better/different than the composition seen in existing benchmarks/datasets like ALFRED [1] or CALVIN [2].

- The purpose of the discussion regarding the 6D-CLIPort agent is unclear. If the intent was to showcase how the environments/tasks from VLMBench could be solved by existing algorithms, or pose a challenge to such techniques, it would have been useful to implement a set of representative baseline algorithms for vision-language manipulation and show the performance on VLMBench. If the intent was to demonstrate a new algorithm for vision language manipulation in general, it would have been useful to compare with other baselines. Fundamentally, the unimodal baseline seems to always be at a disadvantage, as it is intuitive that an RGB+D+language algorithm would perform better. RGB + Depth + Language is a multimodal combination that is not very uncommon in the manipulation field, so several other datasets can also provide this data, and hence it is not really necessary to prove that multimodality is useful and important. It is also not clear how 6DCLIPort performs relative to other best performing VLM techniques in the literature . I think this section needs reframing to really understand the purpose of 6DCLIPort.

- It is also not unclear why the performance of the unimodal baseline (and to some extent, 6D CLIPort without GT) is so extremely low on the benchmark tasks. Also, considering one of the strengths of VLMBench is its ability to generate variations and constraints in the task space, the fact that 6D CLIPort fails so badly at tasks that have high variations (see section 6.3) raises questions about the relevance of this technique.

[1] Mohit Shridhar, Jesse Thomason, Daniel Gordon, Yonatan Bisk, Winson Han, Roozbeh Mottaghi, Luke Zettlemoyer, Dieter Fox, "ALFRED A Benchmark for Interpreting Grounded Instructions for Everyday Tasks"
[2] Oier Mees, Lukas Hermann, Erick Rosete-Beas, Wolfram Burgard, "CALVIN: A Benchmark for Language-Conditioned Policy Learning for Long-Horizon Robot Manipulation Tasks"

**Additional Feedback:**

As discussed in the Weaknesses section, it would be great to clarify 1) the extent of compositionality in VLMBench and whether/how it differs from such properties in ALFRED/CALVIN, and 2) the performance of 6D CLIPort and how it can be localized within the existing body of work for vision language manipulation. That said, I think having a dataset/benchmark simulation that allows for specification of constraints and a good degree of variation in the task/environment is a good contribution.

One more question that might be interesting to discuss: is it possible to extend the types of constraints? For instance, would it be possible to add a constraint that has to do with vicinity to a certain type of object, for example, if the manipulation workspace contained a glass, one natural constraint instruction to specify would be to keep sufficient distance from the glass. Similarly, one might want the robot to speed up or slow down depending on the surroundings. Would it be possible to add such new constraints?

**Clarity:**

The paper is generally well written. The sections regarding AMSolver and VLMBench are easy enough to follow, but as mentioned above, it is hard to gauge the real purpose of 6D CLIPort, how it fits in the existing body of work on vision language manipulation, and why it (and the baselines) perform the way they do.

A couple things that seems to be missing from the paper and would be useful to add:

1. How many classes of objects can VLMBench instantiate and simulate?
2. Does the underlying simulator simulate the physics for the objects as well? i.e., would it be possible to have dynamic objects in the simulation?

**Correctness:**

The evaluation could have been better (through the usage of well known baseline algorithms - currently there seem to be none, and the unimodal baseline is always at a disadvantage making it an unfair comparison), and the associated discussion can be improved greatly.

**Documentation:**

The repo does have a reasonable amount of documentation on how to set up and use the VLMBench framework, along with details on how to perform custom demonstrations or load custom new objects etc.

**Relation To Prior Work:**

The paper does discuss several kinds of manipulation benchmarks from recent years such as RLBench, MetaWorld, CausalWorld. One relevant benchmark/simulator missing from the comparison is CALVIN [1], especially given the compositional focus of CALVIN.

[1] Oier Mees, Lukas Hermann, Erick Rosete-Beas, Wolfram Burgard, "CALVIN: A Benchmark for Language-Conditioned Policy Learning for Long-Horizon Robot Manipulation Tasks"



**Summary And Contributions:**

This paper presents a framework for vision and language manipulation aimed at being a benchmark. VLMBench is a benchmark for vision and language manipulation which contains a manipulator and a tabletop environment. VLMBench provides access to several sensors such as RGB, Depth, segmentation as well as other agent-specific observations such as end effector pose. The agent can be commanded to perform a variety of tasks such as pick and place, stack, drop etc., each task with a potential for varying several parameters such as object type, qualifying the task using constraints such as direction, size, extent etc. Given this information, VLMBench solves the task using a rule based task solver called AMSolver. AMSolver can compute solutions that can adhere to goal/path constraints. Finally, the paper also describes a manipulation agent called 6D-CLIPort, which takes inspiration from CLIPort, and given RGBD observations and a language instruction set for a task, outputs relevant goal positions for the end effector. The authors apply 6D-CLIPort to environments in VLMBench and show that the performance of 6D-CLIPort surpasses that of simple unimodal baselines.

---

> ### Author Response · Authors · 2022-08-17
> **Responses to reviewer DdfD**
>
> We thank the reviewer for appreciating the motivation, presentation, significance, and contributions of this work. Below we hope to address concerns in the review. Feel free to let us know if you have any further questions!
> 1. Re: Comparisons with ALFRED and CALVIN
>
>     We would like to clarify that the compositions of VLMbench differ fundamentally from ALFRED [1] and CALVIN [2]. In ALFRED, the compositions contain finding objects and interacting with them. However, the object interaction in ALFRED does not involve exact manipulation paths and basically assumes the sub-tasks can be automatically finished. For example, the apple will teleport to the robot's hand when the agent produces the action ‘pick up’ before the apple. In CALVIN, the compositions of the long-horizon tasks are different combinations of individual manipulation tasks. However, they do not consider the object variations and the manipulations category. Inside one scene, the object properties, like colors, sizes, and positions, are not changed, and the manipulation tasks are determined by the affordance of objects in the scene. For example, they can only support picking objects, pushing buttons, moving sliders, and moving drawers since these are all affordance in the environment. Moreover, their manipulation demonstrations are collected by human players and cannot be automatically generated. In contrast, VLMbench specifically targets at compositional robotic manipulation tasks specified by various language instructions and can automatically generate the corresponding manipulation demonstrations.
>
> 2. Re: The purpose of the discussion regarding 6D-CLIPort
>
>     In this work, we propose 6D-CLIPort as a baseline algorithm, which is moderately modified from the CLIPort model. We have revised the section to indicate this point. Since the task of (compositional) vision-and-language manipulation is still under-explored, limited methods can deal with multi-modality, high DoFs, compositional reasoning, and significant task variations. We have tried to adapt an imitation learning algorithm [3] and an offline reinforcement learning algorithm [4], but both methods cannot work well on VLMbench. We plan to implement and develop more algorithms on the benchmark, and we welcome the reviewer to provide pointers to some baselines that may be used to solve the VLMbench tasks.
>
> (Continue to the next comment)

---

> > ### Author Response · Authors · 2022-08-17
> > **Responses to reviewer DdfD (continue)**
> >
> > 3. Re: Performance of Unimodal models and 6D-CLIPort
> >
> >     (1). *About the unimodal baseline performance*. Thanks for the question! Since 6D-CLIPort was built upon the codebase of CLIPort, we found that it inherited an issue of the CLIPort codebase resulting in unsuccessfully trained unimodal agents. We fixed the bug and re-trained the agents. The new results can be found in Table 3 of the revision, and below we show the result comparison in the unseen settings.
> >
> >     - (a) The baseline vision-and-language manipulation model 6D-CLIPort performs the best on all tasks, showing the importance of both visual observations and language guidance in VLMbench tasks.
> >
> >     -   (b) The Language-Only agent nearly fails at all the tasks as it is visually blind and thus unable to localize the objects in the 3D space. For "Drawer" tasks, since language instructions can provide the level information (e.g., top, middle, and bottom) and action directions (e.g., open and close), the Language-Only agent has some chance to close the drawer by collisions.
> >
> >     -   (c) Without language guidance, the Vision-Only agent has a significant performance degradation on all tasks. It fails completely on tasks that require more strict pose constraints, including “Drop,” “Shape Sorter,” “Pour,” and “Door.” For other tasks such as Pick&Place", "Stack," “Wipe," and “Drawer,” the Vision-Only agent can succeed in few cases (though with pretty low success rates) by randomly grasping an object in the scene for manipulation because those tasks have a lower variance of the grasping actions and following movements.
> >
> >     More results and analysis can be found in the Experiment section of the revision.
> >
> >     | Agent |Pick&Place | Stack | Drop | Shape Sorter | Pour | Wiper | Door | Drawer |
> >     | :--: | :--:| :--:| :--:| :--:| :--:| :--:| :--:| :--:|
> >     | Language-Only | 0.00 | 0.00 | 0.00 | 0.00 | 0.00 | 0.00 | 0.00|1.04|
> >     | Vision-Only | 9.85 | 1.79 | 0.00 | 0.00 | 0.00 | 20.80 | 0.00|7.29|
> >     | 6D-CLIPort | 27.28 | 18.37 | 6.42 | 12.33 | 1.00 | 21.00 | 5.00|15.63|
> >
> >     (2) *About the 6D-CLIPort performance*. We agree with the reviewer that 6D-CLIPort is not a sufficient method for solving VLMbench tasks, but it is a reasonable baseline for comparison and method development. The compositional vision-and-language manipulation tasks are indeed challenging due to compositional reasoning, large action space with high DoF, and large task variations, and 6D-CLIPort achieves ~15% success rates across all tasks, which is non-trivial. 6D-CLIPort has several advantages. For example, it transfers and leverages the knowledge of the pre-trained CLIP model; therefore, its performance on unseen settings is decent compared to that on seen settings (it is still lower but does not drop dramatically). Moreover, 6D-CLIPort is a general method for all kinds of tasks in VLMbench and does not have specialized designs on model architecture, reward function, and learning algorithm for individual tasks. In other words, the same 6D-CLIPort model architecture is used for all tasks with the same training method. The performance variance on different tasks and the ablation studies show the capabilities and limitations of the 6D-CLIPort model, which we hope can provide indications and insights for researchers to propose better methods to improve the results.
> >
> > 4. > “ How many classes of objects can VLMBench instantiate and simulate?”
> >
> >     Currently, we have five types of objects and 22 classes in total. We list all classes below, and the number behind the classes indicates the instance number inside the class. The object used in each task can be found in Appendix A in the revision.
> >
> >     | Object types | Number of classes | Classes |
> >     | :-----| ----: | :----: |
> >     | Basic model | 3 | cube (1), triangular prism (1), cylinder (1) |
> >     | Special model | 9 | star (1), moon (1), cross (1), flower (1), letter of 't' (1), pencil (1), basket (1), box container(1), shape sorter (1) |
> >     | Planar model | 6 | rectangle (1), circle (1), triangle (1), star (1), cross (1), flower (1) |
> >     | Functional model | 2 | mug (6), sponge (1) |
> >     | Articulated model | 2 | door with one rotatable handle (2), cabinet with three vertical drawers (3) |
> >
> >     With the object mesh and predefined object basic information, like the name of the object, the joint connections of articulated objects, etc., more object models can be seamlessly imported into VLMbench. To add more models, please check the ‘vlm/object_models/save_model.py’ file in the GitHub repository.
> >
> > (Continue to the next comment)

---

> > > ### Author Response · Authors · 2022-08-17
> > > **Responses to reviewer DdfD (continue)**
> > >
> > > 5. > “Does the underlying simulator simulate the physics for the objects as well? i.e., would it be possible to have dynamic objects in the simulation?”
> > >
> > >     Yes, Coppeliasim can support physical simulation very well on rigid objects, even if they are dynamic. For other types of objects like liquid required in the pouring task, it could also be simulated as a set of dynamic particles.
> > >
> > > 6. > “ Is it possible to extend the types of constraints? ”
> > >
> > >     Yes, the constraints can be extended. Our AMSolver considers the construction of manipulation from spatial constraints, including positions and orientations. Therefore, our framework is still feasible for adding constraints in the 3D workspace around the object of interest, like avoiding them by a distance threshold. It can be considered as a special path constraint, which can be customized in the state check function in motion planning. Changing the speed according to the surroundings during the tasks is an interesting situation. To achieve this, we can use the state check function to check whether the arm has reached the specified speed-changing area. Then, the control parameters of the robot arm can be changed to speed up or slow down the response time of joints, which leads to the speed change of the end-effector.
> > >
> > > Reference:
> > >
> > > [1] Shridhar, Mohit, et al. "Alfred: A benchmark for interpreting grounded instructions for everyday tasks." Proceedings of the IEEE/CVF conference on computer vision and pattern recognition. 2020.
> > > [2] Mees, Oier, et al. "CALVIN: A Benchmark for Language-Conditioned Policy Learning for Long-Horizon Robot Manipulation Tasks." IEEE Robotics and Automation Letters (2022).
> > > [3] Stepputtis, Simon, et al. "Language-conditioned imitation learning for robot manipulation tasks." Advances in Neural Information Processing Systems 33 (2020): 13139-13150.
> > > [4] Nair, Suraj, et al. "Learning language-conditioned robot behavior from offline data and crowd-sourced annotation." Conference on Robot Learning. PMLR, 2022.

---

### Official Review · Reviewer_tFzq · 2022-07-28
**Review of paper "VLMbench: A Compositional Benchmark for Vision-and-Language Manipulation"**

**Rating:** 4
**Confidence:** 3

**Strengths:**

- I believe the problem studied here is relevant and important. Vision-based agents executing tasks specified by natural language is of interest to the NeurIPS community and could possibly help with other research including multi-task learning, multi-modal learning, etc
- The paper is overall illustrative, some write-ups could have been clearer though.
- Extending CLIPort to 6 DoF robotic control is interesting, however, the proposed approach does not seem to be working at least on this benchmark.

**Weaknesses:**

Having said those above, I still have some major concerns about the methodology and experiment of this manuscript. I hope the authors could help clarify them in a rebuttal.

- As stated at the end of the introduction, AMSolver is one of the major contributions made by this paper. However, I cannot find enough details about it, thereby making it hard to justify the novelty. Sec. 3 seems to be talking about the grammatical structure of their tasks, which I suppose, should be part of Sec. 4 instead? It is not clear how these task definitions connect to a demonstration generator or a robotic trajectory solver. If my understanding is correct, the authors seem to apply an off-the-shelf solver provided by RLBench to their tasks. If this is the case, I doubt if this can be viewed as a "contribution". The authors are encouraged to elaborate more on the comparison between their proposed AMSolver are other prior arts, ex. the solver in RLBench, BEHAVIOUR, Meta-World, etc.

- In table 1, the authors claim that their benchmark is the only one that includes "automatic 6 DoF grasping". Specifically, they pre-generate some plausible two-fingered grasps on the objects in their benchmark using GPD. However, it's very unclear how could models benefit from these grasping samples when solving their tasks. The authors should at least demonstrate a prototype system that could leverage this data to justify the rationale for making it part of their benchmark.

- VLMBench is claimed to be a "compositional" benchmark. I may be biased but indeed I found it necessary to also include some compositional or OOD generalization tests to further verify the compositional structure of their tasks. This has been a common practice for other compositional benchmarks, including CLEVR, ALFRED, etc. Please kindly remind me if there are already (and make it clearer in the paper).

- The ablations in the experiments indicate that providing ground truth position and orientation could significantly lift the overall performances. However, more details on these model variants should be provided. I can see a total of 6 outputs of the proposed 6DOF-CLIPort. Does "pos" mean x/y/z are already given while the other three will remain predictable, and 'ori' suggest the contrary? It could be hard to parse the results without them. The existing results also could have been more informative other than some numbers, ex. why do ground truth orientation useless than position in most tasks even 6 DOF tasks (Door)? Why does the model still fail even with this privileged information? Some failure mode analysis seems to be necessary to help understand the challenge imposed by this benchmark.

**Additional Feedback:**

See [Weaknesses].

**Clarity:**

Unfortunately, the authors fail to provide enough details on many of their claimed contributions, ex. the trajectory solver and the experiments. The write-ups could have been less glitchy with better proofreading.

**Correctness:**

I found the claims made by the authors supported mainly by their results well, except for the contribution on AMSolver, which I've elaborated on above.

The experiment setting seems riguous. Although I do expect authors could make compositional generalization part of their evaluation protocol.

**Documentation:**

I found the dataset itself well-documented.

**Ethics:**

I didn't find any ethical concerns raised by this benchmark.

**Relation To Prior Work:**

I found the discussion of prior works comprehensive.

**Summary And Contributions:**

Paper introduces VLMbench, a new benchmark for vision-based robot manipulation. Compared to the prior counterparts, the tasks in this benchmark are compositional in terms of action, objects, and their attributes. The benchmark also allows two types of goal definition: goal pose constraint and path pose constraint. To generate demonstrations, the authors propose AMSolver but do not provide many details about it. Finally, some experiments on this benchmark are conducted with a model modified from CLIPort.

---

> ### Author Response · Authors · 2022-08-17
> **Responses to reviewer tFzq**
>
> Thanks for the constructive feedback! We hope to clarify the confusion and answer the questions below. Feel free to let us know if you have any further questions!
> 1. Re: The contribution of AMSolver
>
>     It seems there is a misunderstanding of AMSolver here, and we would like to clarify that *AMSolver is NOT a path planning solver*. AMSolver can be considered as a constructive system to represent a set of manipulation tasks. In general, the manipulation tasks are mainly involved with two parts: objects and constraints, i.e., the manipulation tasks are to manipulate target objects with certain constraints. As shown in Section 3, we can decompose a manipulation task into unit tasks belonging to the templates in AMSolver, where the connection of different unit tasks defines the constraints automatically.
>     So AMSolver represents a manipulation task by multiple unit tasks and automatically samples waypoints or paths for each task step according to the predefined constraints.
>     Then, an off-the-shelf path planning solver (e.g., the one used in RLBench) can be used to verify the feasibility of the sampled waypoints or paths and solve for the entire trajectory. *Therefore, AMSolver and the path planning solver are complementary to each other, while AMSolver introduces modularity and compositionality to the automated creation of manipulation tasks.*
>
>     The solver in BEHAVIOR is used to solve high-level long-horizon task planning problems. It decomposes a long-horizon task (e.g., “making coffee”)  into a sequence of manipulation tasks (e.g., “pick up a coffee mug,” “turn on the coffee machine,” “place the mug under the coffee machine,” etc.), and each of them can be further divided into unit tasks by AMSolver. Meta-World does not provide a solver to plan the trajectories since it focuses on model-free reinforcement learning problems, which requires the agent to learn how to get task completion trajectories.
>
> 2. Re: the significance of grasping samples
>
>     Sorry about the confusion, but we actually use the grasp pose data to train the 6D-CLIPort. During inference, the agent cannot directly acquire the pre-generated grasp samples from the environment. Therefore, the agent needs to estimate the waypoint at each step, and the grasping poses to manipulate objects. Grasping poses are indispensable for a manipulation benchmark as the robots must contact the objects proper for further successful manipulation. Additionally, pre-generated grasping samples associated with object geometry are the prerequisites for automatically generating full-trajectory demonstrations from the object meshes.
>
> 3. Re: compositional or OOD generalization test
>
>     The compositional (or OOD) generalization test can be found in Appendix D.1 of the original submission and Tables 3 & 4 in the revision (we move them into the main paper for better presentation).
>     All tasks in the unseen settings are unseen <color, shape> combinations from an unseen color collection and an unseen shape collection (where the shapes include all object classes and variants).
>     The unseen color collection has five new colors that do not appear during training, including brown, gold, pink, chocolate, and coral.
>     As for the unseen shape collection, it has some overlap with the seen library for the tasks with color variations (but the <color, shape> combinations are always unseen), and is exclusive for the tasks without color variations (e.g., we introduce a new door model with a rotatable handle for the unseen setting of Door tasks). The exact object models used for each task can be found in Appendix A in the revision.
>     So there are mainly three kinds of unseen combinations: <unseen color,unseen shape>, <unseen shape>, and <unseen color,seen shape>.
>     Below we present the success rates of 6D-CLIPort in both seen and unseen settings. From the results, we can see that 6D-CLIPort can generalize to unseen settings to some extent but does not perform as well as in the seen settings.
>
>     |  |Pick&Place | Stack | Drop | Shape Sorter | Pour | Wiper | Door | Drawer |
>     | :--: | :--:| :--:| :--:| :--:| :--:| :--:| :--:| :--:|
>     | Seen | 28.28 | 22.19 | 6.42 | 17.33 | 1,00 | 22.40 | 6.00|22.92|
>     | Unseen | 27.53 | 18.37 | 6.42 | 12.33 | 1.00 | 21.00| 5.00|15.63|
>
> (Continue to the next comment)

---

> > ### Author Response · Authors · 2022-08-17
> > **Responses to reviewer tFzq (continue)**
> >
> > 4. Re: questions about the ablation studies
> >
> >     > ”Does "pos" mean x/y/z are already given while the other three will remain predictable, and 'ori' suggest the contrary?”
> >
> >     Yes,  ‘GT Pos’ means given ground truth x/y/z positions while the other three parameters for 3D orientation are estimated, and ‘GT Ori’ suggests the contrary (known orientation, to estimate 3D position).
> >
> >     > “Why is ground truth orientation useless than position in most tasks even 6 DOF tasks (Door)?”
> >
> > 	One primary reason the GT positions bring more improvement is that it eliminates the difficulties of localizing the target object, one of the main challenges in compositional reasoning. For example, the instruction “place the red cube into the green container” requires the model to localize the correct cube and container, and providing GT positions makes the task much more accessible. Note that all the VLMbench tasks require the ability of compositional reasoning.
> >     Moreover, from the perspective of pose variances, when we divide the task into steps, the orientation of each step has fewer variances than the position in many tasks such as picking, stacking, and wiping.
> >
> >     > “Why does the model still fail even with this privileged information? Some failure mode analysis seems to be necessary to help understand the challenge imposed by this benchmark.”
> >
> >     We agree with the reviewer that error analysis is necessary. In fact, we have shown some failure cases in Appendix D.2 and D.3. As we mentioned in Section 6.4, the ground truth positions and orientations are obtained from the pre-generated waypoints. Therefore, this ground truth information is not the optimal parameter associated with the estimation. In Appendix D.3, we have shown that unmatched positions and orientations can lead to a failure or even an unreachable pose. For the tasks requiring the cooperation of position and orientation, such as pouring and opening the door, we observe that giving partially ground truth may still not guarantee better task completion results. Furthermore, the wrong estimation of positions can lead to task failure even given the ground truth orientation since the tasks in the VLMbench require correct compositional reasoning. These information can be found in Experiment section and Appendix D.3 in the revision.

---

### Author Response · Authors · 2022-08-28
**A kind reminder that the discussion phase is coming to an end**

Thanks to all the reviewers for the appreciations and suggestions of this work! We hope our responses have resolved the confusions and questions in the reviews. Since the discussion phase is ending soon (Aug. 29th), we would like to gently remind the reviewers to take a look at our responses (if they haven’t) and let us know if there are any further comments. Thanks!

---

### Meta-Review · Area_Chair_kHHj · 2022-08-31

**Recommendation:** Accept
**Confidence:** 4

**Metareview:**

The paper presents a benchmark for a domain that is of significant interest to the community, and a complete pipeline is proposed and demonstrated. The work proposes a systematic way to describe tasks in robotic manipulation problems suitable for the dataset/benchmark presented. While reviewers were able to explore the tools presented, clarifications on how to run the tools provided and reproduce results should be clearly present in the documentation for the toolkit itself. The authors are encouraged to consider the strengths and weaknesses identified by reviewers for further clarifying and strengthening the manuscript.

Strengths:
- The problem studied here is relevant and important. Vision-based agents executing tasks specified by natural language is of interest to the NeurIPS community.
- The authors demonstrate a complete pipeline for language-guided robot manipulation research, suitable for getting novel research tasks in this area up and running quickly.
- The paper is well organized and clearly describes the presented benchmark, especially with revisions.
- The presented system allows for specifying a variety of kinds of constraints, making it overall fairly widely applicable.
- Providing automatic 6DoF grasping is a significant improvement of the tool over existing related systems.

Weaknesses:
- Stronger differentiation from existing datasets/benchmarks in the body of the paper would help clarify the contributions of the work.
- The proposed approach does not seem to be working well on CLIPort, which raises questions about the overall relevance of the technique that should be addressed.
- It is not clear from the paper itself how new tasks can be included and evaluated.
- Some newer literature on language-guided robot manipulation should be included.

---

### Decision · Program_Chairs · 2022-09-16

Accept